



**A new global marine gravity model NSOAS24 derived from**
**multi-satellite sea surface slopes**
**Shengjun Zhang[1*], Xu Chen[1], Runsheng Zhou[1], Yongjun Jia[2*]**
**[1] School of Resources and Civil Engineering, Northeastern University, Shenyang, China**
**[2] National Satellite Ocean Application Service, Beijing, China**
**Correspondence to: Shengjun Zhang (zhangshengjun@whu.edu.cn) and Yongjun Jia**
**(jiayongjun@mail.nsoas.org.cn)**
**Abstract**
Judging from the early release of the NSOAS22 model, there were some known issues, such
as boundary connection problems in block-wise solutions and a relatively high noise level. By
solving these problems, a new global marine gravity model NSOAS24 is derived based on sea
surface slopes (SSS) from multi-satellite altimetry missions. Firstly, SSS and along-track deflections
of vertical (DOV) are obtained by retracking, resampling, screening, differentiating, and filtering
procedures on basis of altimeter waveforms and sea surface height measurements. Secondly, DOVs
with a 1'x1' grid interval are further determined by the Green's function method, which applies
directional gradients to constrain the surface, least-square fit to constrain noisy points, and tension
constraints to smooth the field. Finally, the marine gravity anomaly is recovered from the gridded
DOV according to the Laplace Equation. Among the entire processing procedures, accuracy
improvements are expected for NSOAS24 model due to the following changes, e.g., supplementing
recent mission observations and removing ancient mission data, optimizing the step size during the
Green's function method, and special handling in near-shore areas. These optimizations effectively
resolved the known issues of signal aliasing and the "hollow phenomenon" in coastal zones.
Numerical verification was conducted in three experimental areas (Mariana Trench area, Mid-
Atlantic Ridge area, Antarctic area, representing low, mid and high latitude zones) with DTU21, SS
V32.1 and shipborne data. Taking NSOAS22 for contrast, NSOAS24 showed improvements of 1.2,
0.7, 1.0 mGal in 3 test areas by validating with SS V32.1, while declines of 0.6, 0.5, 0.3 mGal, and
0.2, 0.4, 0.3 mGal occurred in STD statistics with DTU21 and shipborne data. Finally, the NSOAS24
was assessed using two sets of shipborne data (the early NCEI dataset and the lately dataset from



JAMTEC, MGDS, FOCD, and SHOM) on global scale. Generally, NSOAS24(6.33 and 4.95 mGal)
showed comparable accuracy level with DTU21 (6.20 and 4.71 mGal) and SS V32.1 (6.40 and 5.53
mGal), and better accuracy than NSOAS22 (6.64 mGal and 5.64 mGal). Besides, the new model is
available at https://doi.org/10.5281/zenodo.12730119 (Zhang et al., 2024).



## 1 Introduction

Satellite altimetry provides highly accurate ocean surface height measurements with respect to certain ellipsoids along corresponding ground tracks (Fu and Cazenave, 2001; Stammer and Cazenave, 2017). Among these altimetry satellites, some have performed geodetic missions (GM) with longer revisit period and denser spatial coverage, which provide the primary data sources for marine gravity recovery. Exact repeat missions (ERM) are also critical in relevant researches according to a relatively lower noise level by averaging nonunique, repetitive cycles (Zhang et al., 2022). Due to new altimetry technology and advanced processing methods, the accuracy of sea surface height (SSH) has increased dramatically over the last decade (Andersen et al., 2023), with a positive influence on marine gravity model construction. The refinement of altimetry-derived marine gravity model has become more obvious due to these recent altimetry missions with dense spatial coverage since 2010, e.g., CryoSat-2, SARAL/AltiKa, Jason-1, Jason-2 and HY-2A GM (Chen et al., 2024). Combining observations from multiple satellites with different orbital inclinations such as 108°, 98°, 92°, and 66° enables a more reliable determination of the marine gravity field (Andersen et al., 2019; Sandwell et al., 2019). In addition to conventional nadir altimeters, synchronized laser beams for obtaining reflected surface height information, two-satellite companion mode, and wide-swath altimetry techniques offer new observations and require effective incorporating strategies for modeling marine gravity field. Furthermore, these advancements provide new opportunities and potentials for recovering refined marine gravity anomalies. Generally, combining multi-frequency and multi-mode altimetry data, especially these observations with higher range accuracy, denser spatial coverage, and diverse track directions, is an effective way of refining marine gravity recovery (Sandwell et al., 2019).

China launched Haiyang-2A (HY-2A) satellite in 2011 and initiated its geodetic mission in 2016 for the purpose of geodetic applications. Multiple previous studies have shown that the HY-2A has consistent accuracy with other conventional altimetry missions (Wan et al., 2020; Zhang et al., 2020; Guo et al., 2022b). Moreover, its followers including HY-2B, HY-2C, and HY-2D were successively launched in 2018, 2020, 2021. Although the HY-2 data cannot serve as the sole input for constructing a 1′x1′ marine gravity anomaly model (Wan et al., 2020; Zhang et al., 2022), the HY-2 series of measurements are extremely valuable for recovering marine gravity anomalies



because of their unique spatial distributions. Currently, several institutions have effectively adopted
HY-2 series data to release regional or global marine gravity models, such as the SCSGA V1. 0 (Zhu
et al., 2020), the NSOAS22 (Zhang et al., 2022), the GMGA1 (Wan et al., 2022), the
SDUST2021GRA (Zhu et al., 2022), and the GMGA2 (Hao et al., 2023). Leaving aside the HY-2
series, the most well-known altimetry-derived marine gravity models are DTU and S&S series,
which are respectively released by the Technical University of Denmark and the Scripps Institution
of Oceanography (SIO), University of California San Diego (UCSD). To some extent, they represent
the highest attainable accuracy (Li et al., 2021; Mohamed et al., 2022). Their latest versions have
been updated to DTU21 and S&S V32.1.
In a previous study of releasing NSOAS22 model, we primarily evaluated the performance of
HY-2 series altimeter data in constructing marine gravity fields and highlighted the role played by
HY-2. However, we found some obvious issues identified in the NSOAS22 through systematic
evaluation. The first and foremost is the boundary connection problem in block-wise solutions,
which lead to a sawtooth-like discontinuity in the final recovered marine gravity signals. Therefore,
this paper aims to address existing issues and to optimize the model-construction steps for the
purpose of constructing refined marine gravity model. These specific improvements contain dataset
filtering and optimization (supplementing recent observations and removing low-quality data), re-
designing the step sizes for solving DOV with Green's functions, and special processing in near-
shore areas. These improvements will be further described in detail in Section 4.
Besides, the remainder of this paper is organized as follows. Section 2 provides a general
description of the involved datasets (altimeter data and shipborne data), as well as the reference
gravity models used for comparison and remove-restore procedure. The theoretical methods for
DOV calculation and gravity anomaly inversion are presented in Section 3. Section 5 evaluates the
altimetry-derived global marine gravity model using the well-known altimetry derived models and
shipborne measurements. Finally, conclusions are given in Section 6, focusing on the 1'x1' global
marine gravity anomaly model named NSOAS24.
**2 Research data**
**2.1 Altimetry data**
The newly accumulated altimetry data has not only provided high-quality SSH observations



but also diverse spatial distributions. For these recent missions, we selected the sensor geophysical
data records (SGDR), which include high-sampling waveforms from the Jason-1, Jason-2, Jason-3,
Cryosat-2, HY-2A, HY-2B, and SARAL/AltiKa. In addition, Jason-1, Jason-2, and SARAL/AltiKa
adopt both ERM and GM data, HY-2A only uses GM data, while HY-2B and Jason-3 only use ERM
data. Cryosat-2 data comprise three modes: low-resolution mode (LRM), synthetic aperture radar
(SAR), and synthetic aperture radar interference (SIN). Taking in account the previously collected
dataset, Geosat observations from both GM and ERM with unique 108° orbital inclination angle,
along with ERS-1 GM, Envisat and TOPEX/Poseidon ERM datasets, were also utilized. Envisat
acquired ERM data for two repeated periods, 30 days and 35 days. The detail information of
involved altimetry data is listed in Table 1.
**Table 1.** Information of altimetry satellites used for deriving gravity field.

| Mission | Satellite | Period | Inclination (°) |
|---|---|---|---|
| Geodetic mission | Jason-1 | 2012.05-2013.06 | 66.00 |
| | Jason-2 | 2017.09-2019.10 | 66.00 |
| | CryoSat-2 | 2010.07-2019.04 | 92.00 |
| | SARAL/AltiKa | 2016.07-2024.01 | 98.55 |
| | HY-2A | 2016.03-2019.06 | 99.30 |
| | Topex/Poseidon | 2002.07-2006.10 | 66.00 |
| | Geosat | 1985.04-1986.11 | 108.10 |
| | ERS-1 | 1994.4-1995.5 | 98.52 |
| Exact repeat mission | Jason-1 | 2008.08-2012.03 | 66.00 |
| | Jason-2 | 2008.07-2017.05 | 66.00 |
| | Jason-3 | 2016.02-2020.07 | 66.00 |
| | SARAL/AltiKa | 2013.03-2016.07 | 98.55 |
| | HY-2B | 2018.11-2023.11 | 99.30 |
| | EnviSat | 2002.05-2012.04 | 98.55 |
| | Topex/Poseidon | 1992.10-2002.06 | 66.00 |
| | Geosat | 1986.12-1990.01 | 108.10 |

Along-track SSS can be considered as vector data, and its magnitude is determined by the time
interval between adjacent ground track points and corresponding SSH variations. Due to different
design of satellite orbital inclinations and ground track orientations (ascending or descending),
along-track SSS capture different signal variations and similar signal variation magnitude with
opposite signs. As shown in Figure 1, satellites with different orbital inclinations exhibit significant





differences in along-track slopes obtained in the Mariana Trench area. Ascending and descending
orbit data both reflect the overall regional trend, exhibiting horizontal symmetry in direction and
being numerically nearly opposite. For instance, the orbital inclination of HY-2A is approximately
99°, allowing it to obtain actual data reaching up to around 81° in high-latitude regions. In contrast,
other altimetry satellites are limited by their designed orbital parameters, such as the Jason series,
which cannot measure data beyond the 66° region. Satellites with near polar orbit have a data
coverage advantage in high-latitude regions. Considering the spatial coverage and orientations, the
calculated slopes should be stored separately based on different orbital inclinations and directions
to ensure the consistency. Consequently, we categorized these satellites in Table 1 into 5 groups
based on their orbital design, as shown in Table 2. For multi-cycle data, these are appended to the
same data file without disrupting temporal continuity, preparing for subsequent segment-based slope
editing steps.

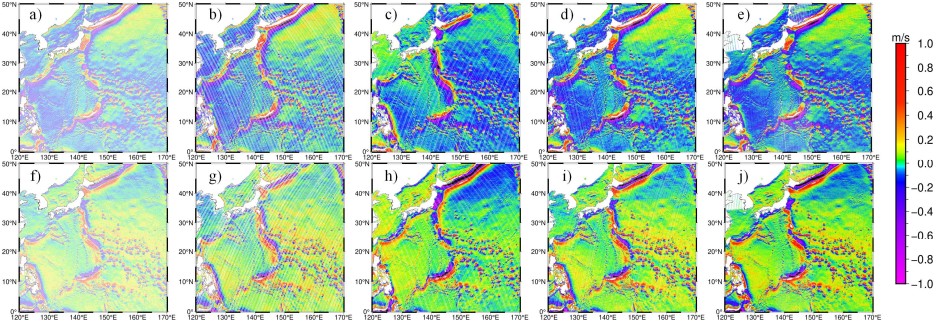

**Figure 1.** Slope plot of satellite ascent/descent at different orbital inclinations (a, b, c, d, e
represent ascending track slopes for HY-2A (99.3°), Geosat (108°), Jason-2 (66°), SARAL/AltiKa
(98.55°), CryoSat-2 (92°) respectively; f, g, h, i, j represent descending track slopes for HY-2A,
Geosat, Jason-2, SARAL/AltiKa, CryoSat-2 respectively).
**Table 2.** Grouping of satellites according to different orbital inclinations.

| i(108°) | ii(98.55°) | iii(66°) | iv(92°) | v(99.3°) |
|---------|-----------|----------|---------|----------|
| Geosat (GM) | Envisat | TOPEX | CryoSat-2(SAR) | HY-2A(GM) |
| Geosat (ERM) | Envisat-P | TOPEX-M | CryoSat-2-P(SAR) | HY-2B(ERM) |
| | SARAL/AltiKa(GM) | Jason-1(GM) | CryoSat-2(SIN) | |
| | SARAL/AltiKa(ERM) | Jason-1(ERM) | CryoSat-2-P(SIN) | |
| | SARAL/AltiKa-F | Jason-2(GM) | CryoSat-2(LRM) | |
| | | Jason-2(ERM) | | |
| | | Jason-3(ERM) | | |



**2.2 Typical gravity models**


To compare and validate the new global marine gravity model, several well-known models
are introduced. Firstly, the latest version of the S&S series model V32.1, which includes both DOV
and gravity anomaly, is used for comparison and validation purposes, hereinafter referred to as the
V32.1. Secondly, the DTU21 gravity anomaly model is introduced for comparison and validation.
Thirdly, the classical EGM2008 comprehensive series model is introduced, which provides the SSH
along with DOT2008A mean dynamic topography model, DOV, and gravity anomaly (Pavlis et al.,
2012). It serves as the reference model in the remove-restore procedure.
**2.3 Shipborne data**
Firstly, a total of 10,740,231 ancient shipborne data points were collected from NCEI (National
Centers for Environmental Information). Secondly, a total of 33,522,351 recent measurements from
four marine institutions with relatively high quality were gathered: FOCD (French Oceanographic
Cruises Directory), JAMTEC (Japan Agency for Marine-Earth Science and Technology), MGDS
(Marine Geoscience Data System), and SHOM (French Naval Hydrographic and Oceanographic
Service). The distribution of shipborne data is illustrated in Figure 2. The NCEI data covers global
oceans more comprehensively, whereas non-NCEI data exhibits dense coverage in the nearby
regions of Japan and in the partial Antarctic seas. Due to inevitable outliers in in-situ data, necessary
data editing was conducted using the triple standard deviation criterion by calculating deviations
with respect to the EGM2008 model. As shown in Figure 2(c), three regions, which are marked in
dashed rectangular and span low, mid, and high-latitude oceans (Area1: 0°-50°N, 120°-170°E;
Area2: 10°-60°N, 310°-360°W; Area3: 50°-80°S, 180°-300°W), were selected as experimental
areas.



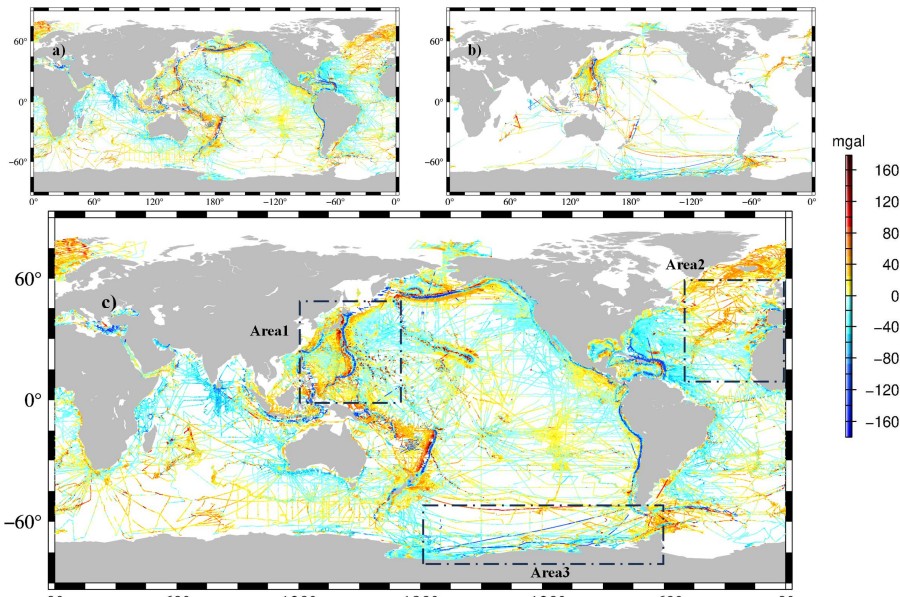

**Figure 2.** Distribution of shipborne data (a: NCEI; b: FOCD, JAMTEC, MGDS, SHOM;

c: Total shipborne data, with three experimental areas highlighted in dashed rectangular).

## 3. Theoretical methodology

### 3.1 Method of along-track DOV calculation

Instead of the derivative of the geoid with respect to the spherical distance, Sandwell et al.

(1997) proposed a method of calculating along-track DOV with two steps. Firstly, geoid slopes were

derived from adjacent geoid heights and corresponding temporal variations. Secondly, the along-

track DOV is computed on basis of geoid slopes by dividing by corresponding satellite orbit

parameter derived velocity. The procedure is summarized in following formula.

$$\varepsilon^\alpha = -\frac{\partial N}{\partial s} = -\frac{\partial N}{\partial t} * \frac{\partial t}{\partial s} = -\frac{\partial N}{\partial t} * \frac{1}{v} \tag{1}$$

The process for determining the linear velocity $v$ is as follows. Given a data point's latitude $\varphi$,

we first convert the geodetic latitude to geocentric latitude $\varphi_c$ by considering the Earth's flattening

$e$. The formula is expressed as follows:

$$\varphi_c = \frac{1-e}{\sqrt{cos^2\varphi+(1-e)^4sin^2\varphi}} \tag{2}$$

Assuming the inclination angle of the satellite's orbit is $\alpha$, the period of the orbit's descending

node is $T$, the regression period is $t$, the distance between adjacent trajectories is $s$, and the



equatorial circumference is $L$, the average angular velocity $w_s$ and synchronous Earth velocity $w_e$
of the satellite's elliptical motion along the orbit can be calculated separately
$$w_s = \frac{2\pi}{T} \tag{3}$$

$$w_e = \frac{w_s t L}{s} \tag{4}$$

Subsequently, the angular velocity components $w_\varphi$ and $w_\lambda$ along the latitude and longitude
directions can be obtained separately
$$w_\varphi = \frac{w_s cos^2\varphi}{(1-e)^2 cos^2\varphi_c}\sqrt{1 - \frac{cos^2\alpha}{cos^2\varphi_c}} \tag{5}$$

$$w_\lambda = \frac{w_s \cos\varphi}{cos^2\varphi_c} - w_e \tag{6}$$

Simple synthesis can obtain the angular velocity $w$ along the orbit
$$w = \sqrt{w_\varphi^2 + w_\lambda^2} \tag{7}$$

Finally, multiply by the radius of the Earth $R$ to obtain the ground linear velocity $v$
$$v = wR \tag{8}$$

**3.2 Method of gridded DOV calculation**
The Green's method proposed by Wessel et al. (1998) restores the along-track DOV to the
gradient direction of the geoid, and subsequently projects it onto the prime (east-west) and
meridional (north-south) components, achieving a similar transformation in the along-track
components (Brammer et al., 1980).
For a linear operator $L$, the output or response under the action of a point source $\delta$ is the Green's
function $G$,
$$LG = \delta \tag{9}$$

where $L$ is taken as the Laplace operator $\nabla^2$,
$$\nabla = i\frac{\partial}{\partial x} + j\frac{\partial}{\partial y} + k\frac{\partial}{\partial z} \tag{10}$$

The Green's function formulation transforms to
$$\nabla^2\phi(x) = \delta(x) \tag{11}$$

The left-hand side of the above equation represents the product of the Laplace operator and
the Green's function formulation, while the right-hand side corresponds to the Dirac delta function.





Solutions that satisfy the Laplace equation are known as harmonic functions, corresponding to cases
where the divergence is zero. The formulation for biharmonic functions is introduced as follows:
$$\nabla^4 \phi(x) = \delta(x) \tag{12}$$

Splines interpolation, whether in one or two dimensions, corresponds physically to enforcing
a thin elastic beam or plate to conform to data constraints. The same interpolation principles apply
to the two-dimensional Green's function formulation as follows:
$$D\nabla^4 \phi(x) - T\nabla^2 \phi(x) = \delta(x) \tag{13}$$

In the equation, $D$ represents stiffness, and $T$ denotes tension factor.
In the discrete case, the following equation holds when there are M reference points within
the region:
$$D\nabla^4 w(x) - T\nabla^2 w(x) = \sum_{j=1}^{M} c_j \, \delta(x - x_j) \tag{14}$$

Wessel et al. (1998) derived the solution $w(x)$ through Fourier transformation as:
$$w(x) = \sum_{j=1}^{M} c_j \, \phi(x - x_j) \tag{15}$$

$$\phi(x) = K_0(p|x|) + \log(p|x|) \tag{16}$$

When there are $N$ known points within the region, the following equation matrix can be
constructed:
$$w_i = \sum_{j=1}^{M} c_j \, \phi(x_i - x_j) \quad i = 1, N \tag{17}$$

Thus,
$$\boldsymbol{w} = \boldsymbol{Gc} \tag{18}$$

The along-track DOV is the projection of the gradient of the geoid along the track direction.
The inverse solution is obtained using the Green's function method, simultaneously applying tension
spline functions to ensure curve smoothness. The fundamental concept is to simulate the geoid field
using a finite number of control points. This approach aims to interpolate and recover the DOV at
all grid points. In discrete conditions, the Green's method formula is shown as equation (14), where
the left-hand side represents selected control points and the right-hand side consists of other known
points with radial basis functions. By iteratively solving from the known points towards the control
points, the radial basis coefficients $c_j$ are determined. This process can be viewed as constructing
the geoid field $\phi$ using finite elements.
Considering that $\phi(x)$ and $w_i$ are scalar fields representing the geoid and their corresponding

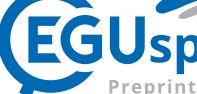

geoid heights, and the actual input data represents the directional derivatives of the geoid,
specifically the along-track DOV vector information. Therefore, introducing the gradient field
$grad\phi(x)$ is formulated as follows in equation (19):
$$\nabla\phi(x) = i\frac{\partial\phi}{\partial x} + j\frac{\partial\phi}{\partial y} + k\frac{\partial\phi}{\partial z} \tag{19}$$

$$s_i = (\nabla w \cdot n)_i = \sum_{j=1}^{M} c_j \nabla\phi(x_i - x_j) \cdot n_i \quad i = 1, N \tag{20}$$

$$\boldsymbol{D} = \sum_{j=1}^{M} c_j \nabla\phi(x_i - x_j) \quad i = 1, N \tag{21}$$

When simultaneously taking the directional derivative in the satellite operation direction $n_i$ on
both sides, $s_i$ represents the along-track DOV vector. $\nabla\phi(x)$ corresponds to the gradient field of the
geoid. Considering the varying quality of data from different satellites, uncertainties $sig$ are
incrementally added to control data quality. Therefore, an equation matrix can be constructed at
reference points:
$$\begin{bmatrix} s_1/sig_1 \\ \vdots \\ s_n/sig_n \end{bmatrix} = \begin{bmatrix} c_1 \\ \vdots \\ c_m \end{bmatrix}^T \begin{pmatrix} 0 & \cdots & D_{x_1-x_m}n_1/sig_1 \\ \vdots & \ddots & \vdots \\ D_{x_n-x_1}n_n/sig_n & \cdots & 0 \end{pmatrix}^T \quad i = 1, N \, j = 1, M \tag{22}$$

After solving for the coefficients $c_j$, the construction of the geoid gradient field is completed.
At any grid point, the geoid gradient $\boldsymbol{D}$ can be determined. Multiplying this gradient by the east-
west and north-south directional vectors yields the DOV components at each grid point.
The Green's function method offers several advantages. Firstly, it innovatively applies
directional gradients rather than SSH to constrain the model surface, in order to enhance stability.
Secondly, it employs least squares fitting instead of exact interpolation, effectively mitigating the
impact of noisy data points. Additionally, by incorporating tension constraints, it facilitates data
smoothing. For moderate data volumes, the Green's function method is superior to traditional finite
difference methods. However, Green's functions also present certain limitations, such as their
inability to handle excessively large datasets, challenges with boundary discontinuities, and
suboptimal performance in near-shore areas. These issues will be discussed and addressed in Section

243 4.

**3.3 Method of deriving gravity anomalies**

The relationship between DOV and gravity disturbances or anomalies can be deduced by the
Laplace equation (Sandwell and Smith 1997). The relationships are established according to the



internal connections among the disturbing potential T , gravity disturbances $\delta$g , gravity anomaly
$\Delta$g , and two directional components of DOV ($\xi$ and $\eta$). Assuming a flat Earth approximation, the
disturbing potential T satisfies the Laplace equation in the given local planar coordinate system (x,
y, z). Then, the relationship between gravity and DOV can be established as the following equation.
$$\frac{\partial \delta g}{\partial z} = -\Upsilon_0 \left( \frac{\partial \xi}{\partial x} + \frac{\partial \eta}{\partial y} \right) \tag{23}$$

Taking the difference between gravity disturbance and gravity anomaly into account, the gravity
anomaly is further calculated according to,
$$\Delta g(x,y) = \delta g(x,y) - \frac{2\Upsilon_0}{R} N(x,y) \tag{24}$$

where $R$ is the average radius of Earth, and $N$ is the geoid height, which can be provided by geo-
potential models. For the detailed computation procedure, please refer to Zhang et al. (2020).
**4 Model construction**
Based on the theories summarized in Section 3, we sequentially calculated along-track SSH,
SSS, along-track DOV, gridded DOV and gridded gravity anomalies from multi-frequency and
multi-mode satellite altimetry data. For the purpose of model construction, a series of joint
processing strategies, e.g., waveform retracking, adding corrections, resampling, data editing,
filtering, as well as the remove-and-restore procedure were necessary.  The specific construction
steps are illustrated in Figure 3.



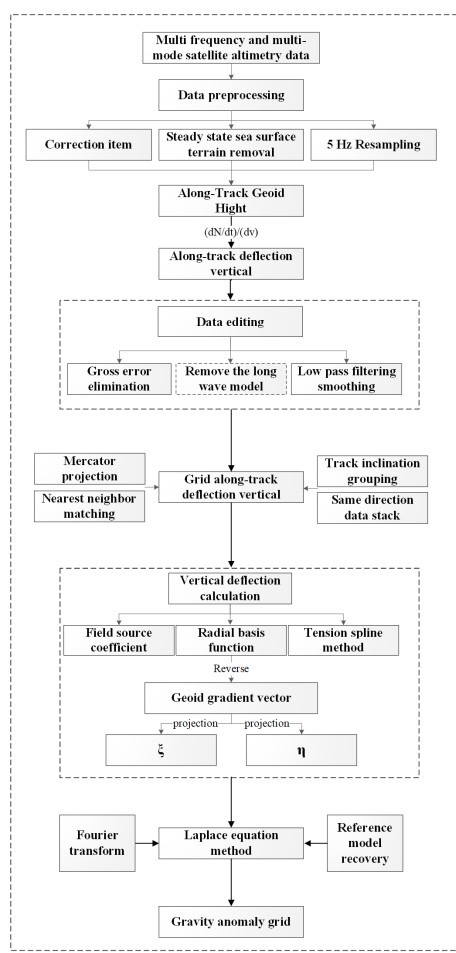

**Figure 3**. Flowchart of constructing marine gravity model from multi-satellite altimeter data.
**4.1 Data preprocessing and slope editing**
Firstly, raw waveforms were retracked using the two-step weighted least-square retracker
(Zhang and Sandwell, 2017), and high-rate observations along profiles were uniformly resampled
into 5 Hz to constrain the noise level and enhance the density of available measurements. Secondly,
along-track SSH measurements were calculated by adding correction items provided in the SDR
products to amend corresponding effects for path delay and geophysical environment. Then the
along-track slopes were calculated, and their accuracy was validated with the EGM2008 model
slopes. If the deviations exceed the setting threshold according to the triple standard deviation
criterion, the data point is considered unreliable and removed. If excessive data segments are edited



out, the entire segment is abandoned to prevent the influence of outliers on subsequent calculations.
Finally, a Parks–McClellan filter was adopted for all these slopes to constrain the amplified high-
frequency noise during the difference procedure.
**4.2 Gridding along-track DOV**
Firstly, along-track velocities corresponding to different satellites were calculated to convert
along-track slopes to along-track DOV. Then along-track residual DOVs were computed by filtering
the EGM2008 geoid heights and corresponding DOT2008A_n180 sea surface topography. Before
gridding, it is necessary to define the objective grid in advance. Considering that the inversion grid
should closely resemble the real earth, a Mercator projection grid was chosen in this study. The
Mercator projection is a cylindrical map projection that preserves angles and is used for a 1'×1'
global grid, with 21,600 grid points in both latitude and longitude directions (The latitude direction
uses the Gudermannian function transformation, while the longitude direction is uniformly divided).
After defining the gridding points, along-track slopes were gridded using a nearest-neighbor
approach. Satellites are categorized based on orbital inclination and ground track orientation, which
ensures that the along-track DOV direction remains consistent and averages potentially redundant
data points in the same direction at grid points, thereby reducing data complexity. Due to the
requirements of the Green's function method regarding region size and data volume, convergence
of multiple vectors with different values at same gridding points with consistent direction can render
the matrix singular. By the way, the averaging step between each category was essential to address
this issue.
As mentioned above, along-track DOVs were mapped to gridding points. Taking the HY-2
group for instance, the gridding process for ascending and descending track segments is illustrated
in Figure 4. Matching is performed using the nearest-neighbor method, and data stacking follows
the principle of consolidating data in the same direction. The specific process is summarized as
follows. (1) Determine the number and position of 1'×1' grid points implemented using the Mercator
projection. (2) Project the geodetic latitude and longitude of input data to Mercator coordinates, and
determine the nearest grid point in the Mercator coordinate system for each data point. (3) Perform
weighted averaging for data in the same direction, and store data from different groups separately.



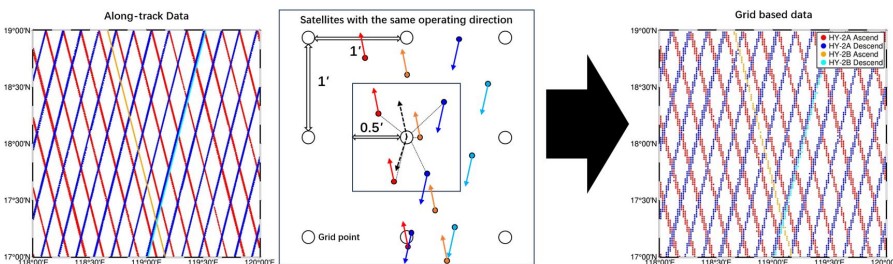

**Figure 4.** Gridding along-track DOVs at grid points (HY-2 group for example)

**4.3 DOV components calculation**

Limited by the computing power of computer and massive gridding points, the DOV components were calculated with block-wise input and output to avoid excessive computational redundancy and matrix singularity. While constructing NSOAS22 model, the tension spline method overlooked the impact of coherence between block-wise regions. This tension spline interpolation is typically suitable for solving small to moderate-sized regions with medium data volumes. However, excessive data can drastically reduce computational efficiency and potentially cause stack overflow issues. Consequently, constraints arising from the distribution of known points may lead to ineffective solving at boundaries and discontinuities between adjacent regions, as illustrated in Figure 5(a). In this study, we proposed a new solution by enlarging computation regions while restricting output to central areas to ensure continuity. Specifically, the inputs were chosen within a 64*64 grid, and the outputs were exclusively limited to the central 32*32 grid. As illustrated in Figure 5(b), the discontinuous effect was eliminated.

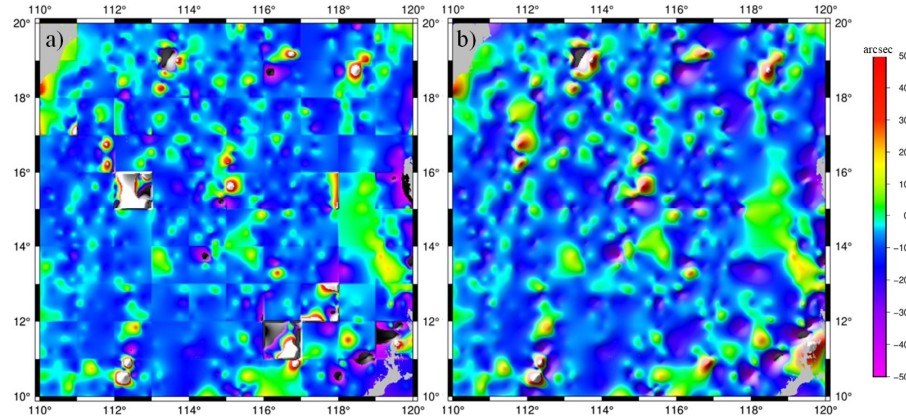

**Figure 5.** Result of spline splicing method for DOV east-west components (a: original; b: new)



### 4.3.1 Step selection

To compute DOV components using the Green's function method, it is necessary to select specific grids as control points for iterative processes. The graphical representation of solving DOV components using the Green's function method is illustrated in Figure 6. Additionally, the tension spline interpolation demonstrates optimal performance when control points are evenly distributed. Leveraging the regularity of the grid, the step size (interval between two control points) is defined for selecting control points. An increased number of control points tends to render the spline curve more rigid, thereby accentuating large fluctuations and noise. Conversely, a reduced number of control points leads to a sparser spline curve that appears smoother, effectively mitigating noise. However, sparse control points may result in an overly simplistic representation of the field. As control points become sparser, the interpolation distance increases, thereby reducing the reliability of the results.

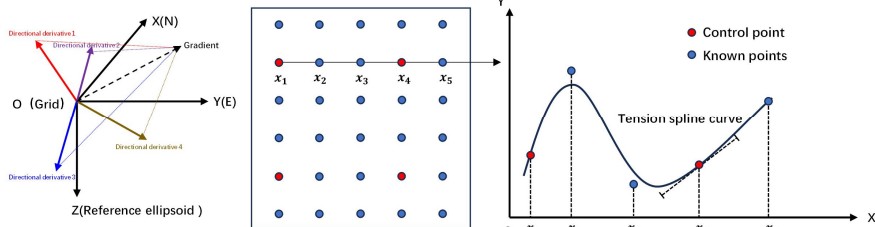

**Figure 6.** Green's function method for solving DOV components

Our computational grid size is 64*64, offering different control point densities based on step sizes: 4096 control points with step size 1, 1024 with step size 2, and 441 with step size 3. Larger step sizes lead to fewer control points, which may not adequately represent the region. Step size 1 results in excessive noise, affecting signal continuity and computational efficiency. Hence, step sizes 2 and 3 are under consideration in our study for balancing detail and computational feasibility.

In experimental area 1 in Figure 2(c), the residual DOV for step sizes 2 and 3 is shown in Figure 7. The figure demonstrates that with a step size of 2, noticeable noise artifacts are introduced, particularly impacting the east-west components. In contrast, using a step size of 3 results in smoother outcomes, exhibiting clearer distribution characteristics of the DOV components. The reduction of noise is particularly effective in specific areas like near-shore regions and islands.



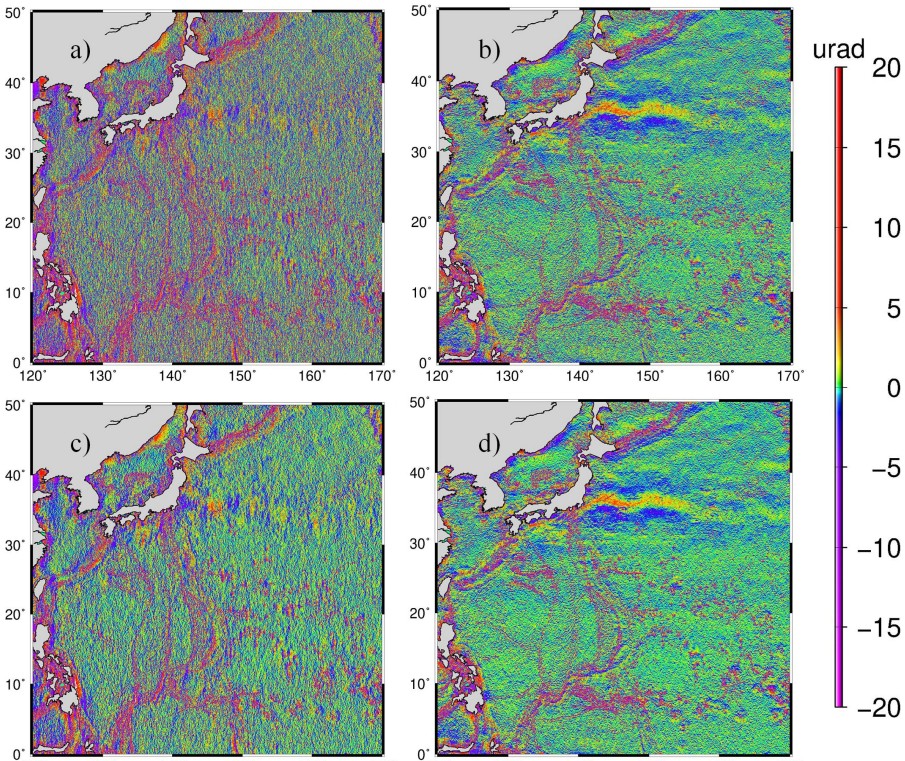

**Figure 7.** Residual results of DOV components difference for different step size selections (a: east-west component at 2 steps; b: north-south component at 2 steps; c: east-west component at 3 steps; d: north-south component at 3 steps)

Then we analyzed the distribution of noise under different step sizes. The V32.1 serves as an verification model, against which the DOV results obtained with a step size of 2 are subtracted. The standard deviation is 3.19 μrad for the east-west component and 2.02 μrad for the north-south component. Setting a threshold based on the triple standard deviation criterion, the primary noise locations are depicted in Figure 8(a) and (b). There are 125,456 noise points in the east-west component, accounting for 1.20% of the entire region, and 122,976 noise points in the north-south component, making up 1.19% of the total area. After removing these noise points, the standard deviations reduce to 2.45 μrad for the east-west component and 1.36 μrad for the north-south component. With a step size of 3, the standard deviations are respectively 2.37 μrad for the east-west component and 1.75 μrad for the north-south component. Identifying based on the triple standard deviation criterion, the primary noise locations are shown in Figure 8(c) and (d). There are




77,904 noise points in the east-west component, accounting for 0.75% of the entire region, and
105,923 noise points in the north-south component, comprising 1.02% of the total area. After
removing outliers, the standard deviations decrease to 1.84 µrad for the east-west component and
1.20 µrad for the north-south component.

It is evident that the noise in the east-west component is noticeably reduced with a step size of

3 compared to that with a step size of 2. Moreover, scattered noise points in open ocean areas are
massively eliminated. This is to say, the selection of step size significantly influences both the
distribution and magnitude of noise points. Considering on larger step size's advantages in enhanced
computational efficiency, reduced matrix complexity, and lower mitigate noise, we finally selected
step size 3 for acquiring controlling points.

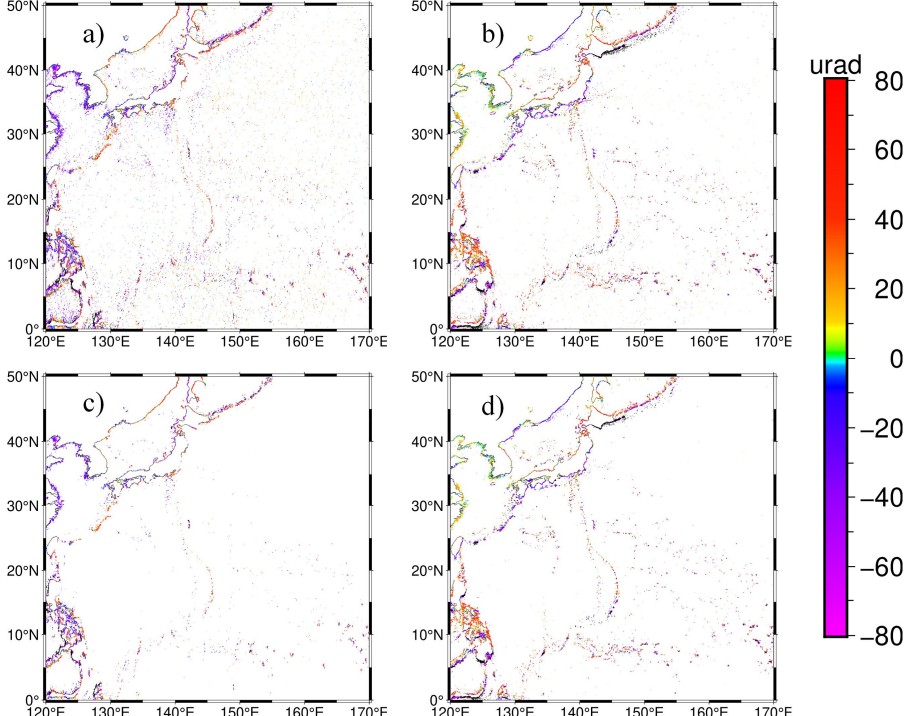


**Figure 8.** Noise analysis at different step sizes (a: east-west component at step size 2; b:
north-south component at step size 2; c: east-west component at step size 3; d: north-south
component at step size 3)

In addition, comparisons between step sizes were conducted in two other experimental areas,

and the statistical results are presented in Table 3. It's interesting that Experimental area 3 exhibits
distinctive characteristics. Satellites with lower inclinations, such as the Topex/Poseidon and Jason



series, are unable to provide observations beyond 66°, resulting in a noticeable decline in DOV
quality in high-latitude regions.
**Table 3.** Statistics of DOV components with respect to V32.1 for different step sizes (unit: µrad)

| Area | Step size | DOV components | Max | Min | Mean | STD |
|------|-----------|----------------|------|------|------|------|
| 1 | 2 | East-west | 623.07 | -610.62 | -0.02 | 3.19 |
|   | 3 | East-west | 258.74 | -393.84 | -0.02 | 2.37 |
|   | 2 | North-south | 613.82 | -614.79 | 0.01 | 2.02 |
|   | 3 | North-south | 388.40 | -401.70 | 0.01 | 1.75 |
| 2 | 2 | East-west | 326.62 | -327.40 | -0.03 | 2.40 |
|   | 3 | East-west | 628.80 | -286.61 | -0.03 | 1.80 |
|   | 2 | North-south | 327.37 | -328.91 | 0.00 | 1.50 |
|   | 3 | North-south | 400.27 | -584.03 | 0.00 | 1.39 |
| 3 | 2 | East-west | 634.40 | -639.41 | 0.11 | 5.41 |
|   | 3 | East-west | 518.80 | -644.39 | 0.09 | 4.34 |
|   | 2 | North-south | 636.89 | -634.96 | -0.09 | 4.61 |
|   | 3 | North-south | 620.09 | -522.40 | -0.10 | 3.74 |

**4.3.2 Special processing in near-shore areas**
Along the coastline, SSH measurements are typically available only on the ocean side, while
grid points over land are default values and posing computational challenges. As illustrated in Figure
8, increasing the step size effectively reduced a considerable number of noise points over the open
sea, while the remaining noise points majorly concentrated in near-shore areas. To demonstrate the
effect of special processing in near-shore areas, we chose the China sea and its adjacent waters
(100°-140°E, 0°-40°N) as the experimental area. This area is densely distributed with islands and
reefs, involving typical categories of coastal regions. Based on the calculated residual DOV with
respect to V32.1, we distinguished noise points where the absolute deviation exceeds 20 µrad. The
distribution of noise points near the coastlines is more pronounced, as shown in Figure 9. The east-
west component and north-south component noise points account for 0.27% and 0.09% of the total
grid points in the region, respectively. It is evident that larger noise points are more prevalent in the
anomalous computation of the east-west component. Therefore, special treatment is required in
near-shore areas to mitigate the concentrated occurrence of noise. As previously mentioned, the
Green's function method operates within a 64*64 grid area. When handling near-shore regions, the
grids over land lack data, with controlling points only available on the ocean side. Thus, the actual
data boundary is at the coastline, but not at the edges of the 64*64 grid. These mixed zones directly
cause boundary effects that hinder matrix convergence. Expanding the computation area is not a
feasible solution because even with an enlarged area, there are no effective data points over land to

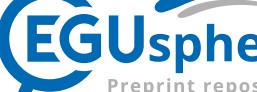



provide constraints. Solutions without constraints typically exhibit lower reliability, contributing
significantly to the observed noise in coastal areas. Figure 10 further gives these differences between
calculation and V32.1 over land-influenced 64*64 grid areas, showing the approximate outline of
the block-wise rectangular computational regions in finer detail.  The influential grid points in near-
shore areas account for 10% of the total grid points. Additionally, there are 30% of grid points over
land within the influential region, indicating a significant proportion of near-shore grid points.

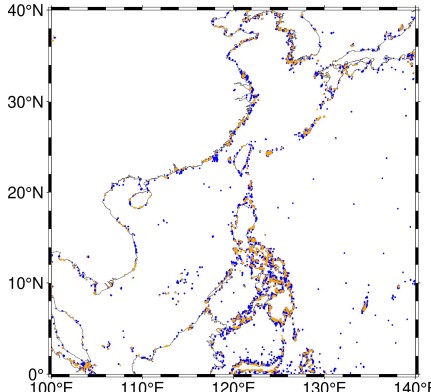

**Figure 9.** The main locations of noise distribution in the near-shore area
(east-west component in orange, north-south component in blue)

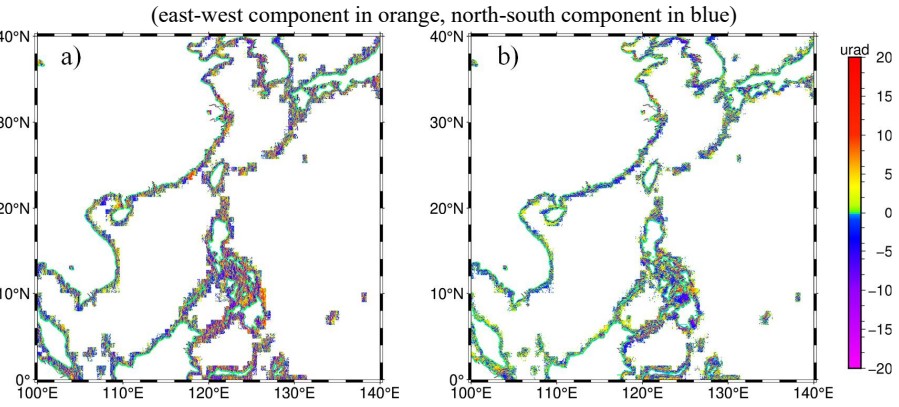

**Figure 10.** Location of distribution of nearshore areas disturbed by continental regions
(a: east-west component; b: north-south component)
To constrain this boundary effect, special processing steps were implemented. A continental
mask was applied to identify controlling points over land, which were assigned a value of 0 and
treated as known points. Moreover, these points were assigned relatively huge uncertainties to
minimize their weight. This approach effectively mitigated boundary effects, thereby controlling
data divergence and improving the reliability of computations in these land-influenced regions.



Figure 11 illustrates the difference in nearshore points before and after processing. Following the
adjustments, there is almost no change on the seaward side. Whereas on the landward side, the
standard deviation shows a difference of 1.67 µrad in the east-west component and 1.47 µrad in the
north-south component, with a maximum difference of around 60 µrad. This indicates that this
special processing effectively suppressed the occurrence of large noise points near the coastlines.

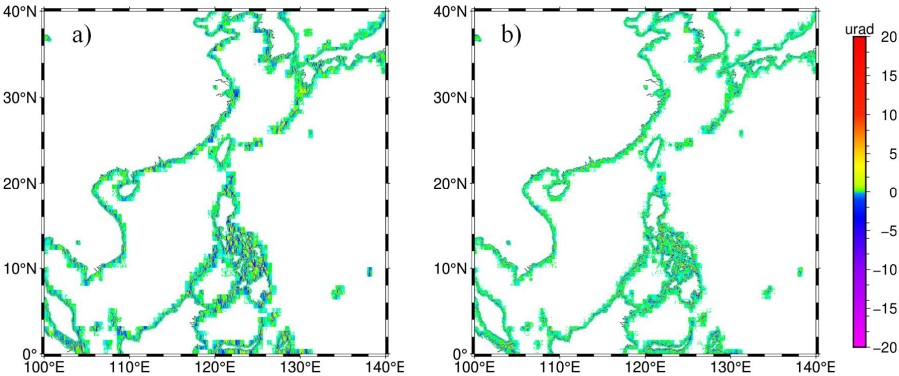

**Figure 11.** Difference in results in the nearshore area before and after the special processing
(a: east-west component; b: north-south component)
Statistical analysis was also conducted in three experimental areas, and the results are listed in
Table 4. The first and foremost is that the nearshore constraint effectively reduced the magnitude of
maximum and minimum deviations. Especially in areas 1 and 2, maximum and minimum values
were notably declined, indicating an effective constraint on the occurrence of large noise spikes.
Moreover, similarly using a deviation threshold of 20 µrad for identifying noise points, the overall
noise ratios decreased by 17.6% following this optimization effort.
**Table 4.** Statistics on the difference with respect to V32.1 with or without nearshore constraint
(unit: µrad)

| Area | Near-shore constraints | DOV components | Max | Min | Mean | STD |
|---|---|---|---|---|---|---|
| 1 | Yes | East-west | 110.52 | -96.43 | -0.03 | 2.42 |
| | No | East-west | 258.74 | -393.84 | -0.02 | 2.37 |
| | Yes | North-south | 68.41 | -87.66 | 0.02 | 1.76 |
| | No | North-south | 388.40 | -401.70 | 0.01 | 1.75 |
| 2 | Yes | East-west | 95.22 | -75.06 | -0.03 | 1.77 |
| | No | East-west | 628.80 | -286.61 | -0.03 | 1.80 |
| | Yes | North-south | 81.95 | -70.86 | 0.01 | 1.28 |
| | No | North-south | 400.27 | -584.03 | 0.00 | 1.39 |
| 3 | Yes | East-west | 447.94 | -644.39 | 0.09 | 4.35 |
| | No | East-west | 518.80 | -644.39 | 0.09 | 4.34 |
| | Yes | North-south | 620.09 | -461.79 | -0.10 | 3.81 |
| | No | North-south | 620.09 | -522.40 | -0.10 | 3.74 |





### 4.3.3 Remove ERS-1 data

To evaluate the contribution of each individual mission to multi-satellite altimetry derived DOV, each satellite (SARAL/AltiKa, EnviSat, HY-2A/B, Geosat, and ERS-1) was sequentially removed within the China sea and its adjacent waters (100°-140°E, 0°-40°N). Median Absolute Deviations (MAD) of the east-west and north-south components along latitude were computed, with the NSOAS24 DOV without data removal used as a comparison. Land-influenced zero values were excluded during this experiment. The results were presented in Figure 12, which illustrates that SARAL/AltiKa provides the most reliable data and the largest contribution. HY-2 also significantly influences the DOV, resulting in discrepancies exceeding 2.5 μrad in the east-west component and ranging from 1 to 1.5 μrad in the north-south component. ERS-1 and Geosat have a relatively minor contribution, causing differences of less than 1.5 μrad and 1 μrad respectively in the east-west and north-south components. This also suggests that their signals overlap to a greater extent with other satellites.

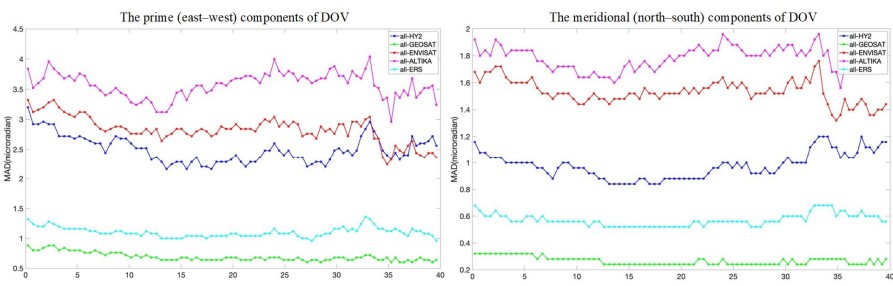

**Figure 12.** Difference in median absolute deviations between NSOAS24 DOV and DOV in the absence of certain mission (The greater the difference, the larger the influence).

Additionally, DOV components were calculated for several single satellite mission, and the MAD between them and V32.1 in latitude direction was compared. As shown in Figure 13, the MAD values are consistently small for HY-2, ENVISAT, and SARAL/AltiKa. However, the data from Geosat and ERS-1 exhibit significant deviations, suggesting considerably higher noise levels.



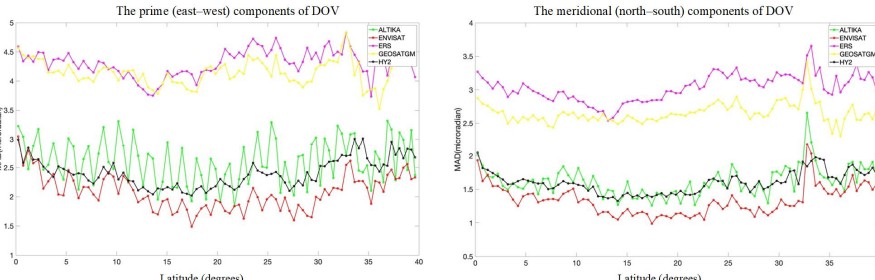


**Figure 13.** Difference in median absolute deviations between V32.1 DOV and the single satellite

solution (The smaller the difference, the better the DOV solution).

Due to being in the early stages of satellite altimetry, Geosat and ERS-1 may suffer from

inherent ranging errors and orbit determination issues that could lead to degraded data quality.
Considering the vast amount of observations accumulated in recent decades, it is worthwhile to
consider removing low-quality and redundant data. For Geosat, its extensive accumulated data
volume and dense coverage in high-latitude region, coupled with its unique 108° orbital inclination,
make it a distinct group of observations with independent direction. Therefore, we have chosen to
temporarily retain Geosat data in the NSOAS24 model construction. ERS-1 has also accumulated a
significant amount of data. However, within the same directional group in Table 2, SARAL/AltiKa
and Envisat share a substantial number of grid points that overlap completely with ERS-1
(accounting for 30.7% of overlap). During the gridding process, these overlapping data points were
stacked. In other words, 30.7% of ERS-1's data can be entirely replaced by higher-precision data
from SARAL/AltiKa and Envisat. From the perspective of controlling points, it is noteworthy that
control points in all directions exhibit a duplication rate exceeding 95%. Therefore, with adequate
data coverage, multidirectional and high-quality precise slope data are required. Considering the
previously identified poor performance and high replaceability, ERS-1 data has been ultimately
removed in NSOAS24 model construction.
**4.4 Gravity anomaly inversion procedure**

Based on the DOV components at grid points, the residual gravity anomalies were calculated

using the FFT method according to the Laplace Equation derived relationship, and the results were
shown in Figure 14. Finally, a global marine gravity model over a range of 80°S-80°N with a 1′×1′
grid interval, named NSOAS24, was constructed after restoring the removed reference model, as



shown in Figure 15.

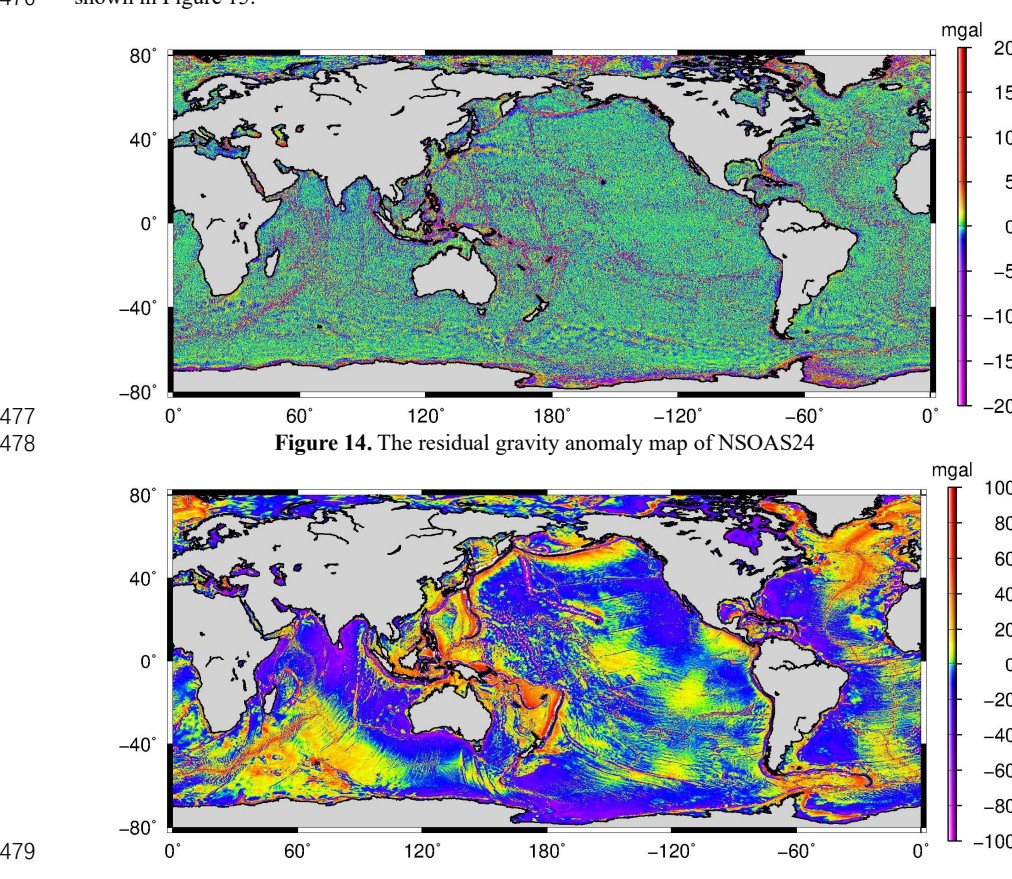

**Figure 14.** The residual gravity anomaly map of NSOAS24

**Figure 15.** The gravity anomaly map of NSOAS24

## 5 Gravity anomaly results

### 5.1 Comparison with V32.1 and DTU21

Firstly, the reliability of NSOAS24 was validated using altimetry-derived models, e.g., DTU21
and V32.1, with statistical results summarized in Table 5. In Area 1 with relatively complex seafloor
terrains, which includes the Mariana Trench, seamount chains, and numerous nearshore areas,
NSOAS24 shows improvements of 0.6 mGal and 1.2 mGal  over its predecessor (NSOAS22),
compared to DTU21 and V32.1, respectively. In the predominantly open sea Area 2, NSOAS24
demonstrates enhancements of 0.5 mGal and 0.7 mGal  over NSOAS22, compared to DTU21 and
V32.1, separately. Area 3 shows a 0.3 mGal improvement for NSOAS24 over NSOAS22, compared
to DTU21, and a 1.0 mGal improvement compared to V32.1.





**Table 5.** Statistics of NSOAS24 and its predecessor against DTU21 and V32.1 (unit: mGal)

| Area | Model | Max | Min | Mean | STD |
|------|-------|-----|-----|------|-----|
| 1 | NSOAS22-DTU21 | 202.64 | -196.75 | -0.09 | 3.56 |
| | NSOAS24-DUT21 | 238.23 | -255.97 | 0.02 | 2.93 |
| | NSOAS22-V32.1 | 167.32 | -196.69 | -0.13 | 3.15 |
| | NSOAS24-V32.1 | 91.36 | -243.28 | -0.03 | 1.97 |
| 2 | NSOAS22-DTU21 | 71.89 | -163.61 | -0.07 | 1.96 |
| | NSOAS24-DUT21 | 104.50 | -72.45 | -0.03 | 1.46 |
| | NSOAS22-V32.1 | 63.64 | -159.40 | -0.05 | 1.95 |
| | NSOAS24-V32.1 | 109.01 | -101.60 | -0.01 | 1.23 |
| 3 | NSOAS22-DTU21 | 90.40 | -167.89 | 0.02 | 6.32 |
| | NSOAS24-DUT21 | 195.52 | -223.07 | 0.02 | 6.01 |
| | NSOAS22-V32.1 | 329.41 | -195.06 | -0.08 | 4.63 |
| | NSOAS24-V32.1 | 305.43 | -188.36 | -0.08 | 3.61 |

**5.2 Comparison with shipborne gravity data**
The distribution of shipborne data and corresponding gravity anomalies of NSOAS24 model
in three experimental areas are illustrated in Figure 16. In Area 1, NCEI data show relatively even
distribution, while JAMTEC data are concentrated near Japan with dense nearshore coverage. In
Area 2, NCEI data are involved within entire region, while FOCD and SHOM data are primarily
concentrated along the Mid-Atlantic Ridge. In Area 3, NCEI data are sparse, with fewer
observations, whereas MGDS data are more evenly distributed and voluminous. Statistical
comparisons are presented in Table 6. The analysis highlights that NSOAS24 significantly improves
accuracy compared to NSOAS22. Furthermore, NSOAS24 demonstrates accuracy comparable to
DTU21 and V32.1, and outperforms V32.1 in the high-latitude polar regions.
Finally, these models were validated using worldwide distributed shipborne data. The accuracy
of each model was assessed using two sets of shipborne data: the early NCEI dataset and the recent
high-quality dataset from JAMTEC, MGDS, FOCD, and SHOM. The results are summarized in
Table 7. In general, NSOAS24 demonstrates accuracy comparable to DTU21 and V32.1. Compared
to its predecessor, NSOAS24 shows a steady improvement in accuracy, with a declination of ~0.7
mGal in standard deviations when compared with recent non-NCEI shipborne data.





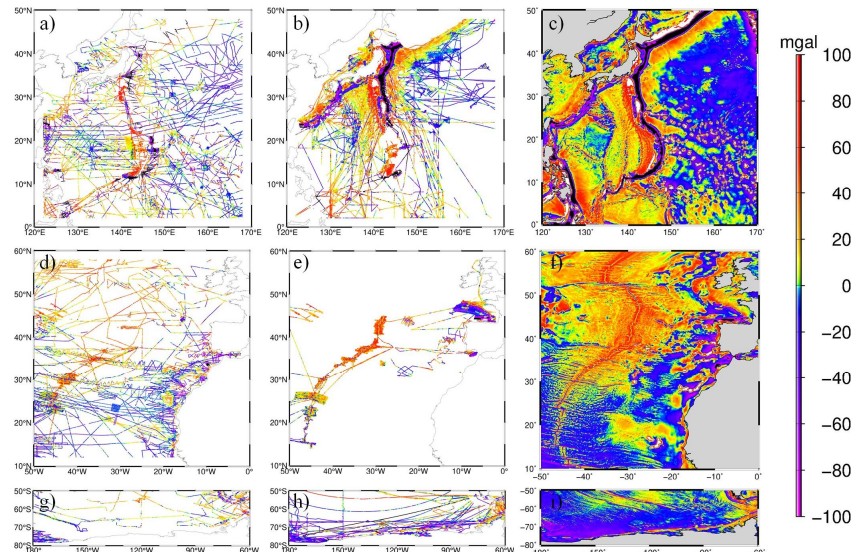


**Figure 16.** Distribution of NCEI and non-NCEI shipborne data, and recovered gravity anomalies

**Table 6.** Statistics on differences between altimeter-derived models and shipborne gravity data (unit: mGal)

| Area | Model | Ship-borne data | Max | Min | Mean | STD | Ship-borne data | Max | Min | Mean | STD |
|------|-------|-----------------|-----|-----|------|-----|-----------------|-----|-----|------|-----|
| 1 | NSOAS22 | NCEI | 55.82 | -45.38 | -0.52 | 6.14 | JAMTEC | 40.20 | -42.62 | 0.97 | 5.18 |
| | NSOAS24 | | 37.07 | -41.06 | -0.68 | 5.60 | | 35.15 | -42.88 | 1.12 | 4.97 |
| | DTU21 | | 36.09 | -42.72 | -0.72 | 5.12 | | 24.90 | -26.20 | 0.56 | 4.37 |
| | V32.1 | | 54.68 | -68.25 | -0.68 | 5.07 | | 57.91 | -66.35 | 0.74 | 4.99 |
| | EGM2008 | | 15.00 | -15.00 | -0.61 | 5.70 | | 15.00 | -15.00 | 0.52 | 4.93 |
| 2 | NSOAS22 | NCEI | 33.02 | -29.06 | 3.07 | 7.28 | FOCD SHOM | 29.74 | -31.69 | 2.93 | 6.95 |
| | NSOAS24 | | 27.35 | -27.61 | 3.24 | 7.21 | | 26.67 | -25.17 | 3.17 | 6.60 |
| | DTU21 | | 23.96 | -22.73 | 3.16 | 7.17 | | 22.14 | -18.70 | 3.14 | 6.48 |
| | V32.1 | | 36.84 | -26.69 | 3.19 | 7.19 | | 29.45 | -19.20 | 3.16 | 6.45 |
| | EGM2008 | | 15.00 | 15.00 | 2.87 | 7.13 | | 15.00 | 15.00 | 2.78 | 6.74 |
| 3 | NSOAS22 | NCEI | 35.16 | -46.12 | 2.54 | 6.40 | MGDS | 39.72 | -43.68 | -0.10 | 6.18 |
| | NSOAS24 | | 189.58 | -38.36 | 2.56 | 6.21 | | 44.94 | -68.45 | -0.09 | 5.92 |
| | DTU21 | | 23.28 | -41.36 | 3.20 | 5.79 | | 44.68 | -58.59 | 0.16 | 5.83 |
| | V32.1 | | 279.57 | -142.13 | 2.64 | 7.79 | | 235.69 | -114.62 | 0.41 | 8.68 |
| | EGM2008 | | 15.00 | 15.00 | 2.42 | 6.28 | | 15.00 | 15.00 | -0.08 | 6.19 |


**Table 7.** Verifications with globally distributed shipborne data (unit: mGal)

| Model | Ship-borne data and number(pcs) | Max | Min | Mean | STD | Ship-borne data and number(pcs) | Max | Min | Mean | STD |
|-------|--------------------------------|-----|-----|------|-----|--------------------------------|-----|-----|------|-----|
| NSOAS22 | NCEI (10740231) | 56.39 | -67.77 | 1.48 | 6.64 | JAMTEC | 48.46 | -48.02 | 1.00 | 5.64 |
| NSOAS24 | | 183.63 | -134.00 | 1.49 | 6.33 | MGDS | 48.08 | -156.23 | 1.01 | 4.95 |
| DTU21 | | 46.37 | -57.59 | 1.34 | 6.20 | FOCD | 44.68 | -81.73 | 0.71 | 4.71 |
| V32.1 | | 279.59 | -193.98 | 1.41 | 6.40 | SHOM | 297.00 | -114.62 | 0.86 | 5.53 |
| EGM2008 | | 15.00 | -15.00 | 1.24 | 6.33 | (33522351) | 15.00 | -15.00 | 0.67 | 5.20 |

**6 Conclusions**
Based on our global marine gravity model construction experience in NSOAS22, we initially





optimized the dataset by incorporating recent observations and excluding highly substitutable ERS-
1 data. Then, multi-satellite datasets were uniformly prepared for constructing a new global marine
gravity model. During the processing, satellites with different orbital inclinations were firstly
grouped into 5 categories. For multi-cycle ERM data, they were appended to the same data file in a
way that preserves the temporal continuity of the data without disruption. Secondly, raw waveforms
were retracked using the two-step weighted least-square retracker, and high-rate observations along
profiles were uniformly resampled into 5 Hz to enhance the density of available measurements.
Thirdly, pre-processing and slope editing were applied to the SSH measurement data to remove
outliers, and the Parks–McClellan filter was used to constrain the amplified high-frequency noise
during the differencing procedure. Fourthly, the residual along-track DOV was calculated from
slopes by dividing by corresponding along-track velocities and introducing EGM2008 as a reference
model. Fifthly, gridded DOV were determined from along-track DOV by the Green's function
method. Finally, a global marine gravity model was constructed after FFT and corresponding inverse
transform, restoring the removed reference model.

Comparing with the predecessor NSOAS22, several optimizations and improvements were

implemented during the entire processing procedures for building NSOAS24. (1) Employing block-
based input and output, calculations were executed with a 64*64 grid input and output the central
32*32 grid. This improvement effectively resolved poor accuracy issues at boundaries and
eliminated discontinuities between adjacent regions. (2) Utilizing the Green's function method to
solve the DOV components, we increased the step size from 2 to 3 for selecting grid points as control
points for iteration. This optimization aimed to enhance computational efficiency, reduce matrix
complexity, and achieve noise smoothing effects. (3) We implemented specialized processing in
coastal regions by incorporating a continental mask. The identified land points were assigned a
default value with huge uncertainty to mitigate their weight. This approach effectively suppressed
boundary effects near coastlines and controlled data divergence.

The new NSOAS24 model was firstly validated with well-known altimetry derived models.

Comparisons were made in three experimental areas (Low-latitude, Mariana Trench area; mid-
latitude: Mid-Atlantic Ridge area. High-latitude, Antarctic area) against the DTU21 and V32.1.
Compared to the predecessor NSOAS22, NSOAS24 showed improvements of 0.6 mGal, 0.5 mGal,



0.3 mGal, and 1.2 mGal, 0.7 mGal, 1.0 mGal, respectively.  Next, we utilized two sets of shipborne
data to verify the new model: the earlier NCEI dataset and the recent non-NCEI dataset collected
from JAMTEC, MGDS, FOCD, SHOM. NSOAS24 also demonstrated a steady improvement in
accuracy compared to NSOAS22. Finally, on a global scale, we validated NSOAS24 (6.33 mGal
and 4.95 mGal) using the NCEI dataset and the combined dataset from JAMTEC, MGDS, FOCD,
and SHOM (6.20 mGal and 4.71 mGal for DTU21; 6.40 mGal and 5.53 mGal for V32.1).
NSOAS24's accuracy was comparable to DTU21 and V32.1, with a notable improvement over
NSOAS22 (6.64 mGal and 5.64 mGal). It is worth mentioning that NSOAS24 showed a decline in
standard deviations of around 0.7 mGal compared to NSOAS22 when comparing with non-NCEI
data. In conclusion, validations with both altimetry-derived models and shipborne data proved the
effectiveness of optimizations and reliability of the NSOAS24 model.

**Author contributions**

SZ and RZ contributed to the development of the global marine gravity anomaly model. Writing of
the original draft was undertaken by XC and SZ, and YJ contributed to review and editing. All
authors checked and gave related comments for this work.

**Data availability**

The global marine gravity anomaly model, NSOAS24, is available at the ZENODO repository,
https://doi.org/10.5281/zenodo.12730119 (Zhang et al., 2024). The dataset includes global marine
gravity anomalies in NetCDF file fortmat.

**Competing interests**

The contact author has declared that none of the authors has any competing interests.

**Acknowledgements**

We are very grateful to AVISO for providing the altimeter data, and NCEI, JAMTEC, MGDS,
FOCD, SHOM for providing shipborne gravity. We are also thankful to SIO and DTU for their
published altimetry derived gravity models. Thanks to ICGEM for providing earth gravity models.



**Financial support**
This study was supported by the National Nature Science Foundation of China, grant number

421932513.

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
