# Peer review of "A new global marine gravity model NSOAS24 derived from"

_EGUsphere, 2024_

## Author Response (AR1)

**Reply to Comments**

**Table of Contents**

**1 Reviewer #1**

(Comments to Author):

The manuscript presents the development of a new global marine gravity model, NSOAS24, which utilizes updated data from multiple nadir satellite altimetry missions, including HY-2. The manuscripts address limitations in a previous model, NSOAS22, also developed by the lead author. The authors proposed several technical improvements, including boundary inconsistencies, dataset filtering, near-shore area processing, and re-designing step sizes. The NSOAS24 model has comparable performance compared to the Sandwell & Smith model and DTU model. The work is relevant and aligns well with the ongoing need for high-resolution marine gravity models for applications in geophysics, oceanography, and remote sensing. My primary concern is that the authors compare NSOAS24 against the Sandwell & Smith V32.1 and DTU21 models, which use a shorter timespan of nadir satellite radar altimetry data. Including an explanation of this discrepancy in the manuscript would enhance clarity. Overall, I see no major technical issues and recommend a minor revision.

Dear reviewer:

The author's team would like to thank you very much for taking your valuable time to review the paper and for providing very valuable feedback and suggestions.

NSOAS24, V32.1 and DTU21 have slight difference in the input multi-satellite altimeter dataset. For instance, the combination of HY-2 data is unique for NSOAS24, while the other two models also used extra sentinel-3 data. Therefore, we did not analyze the specific differences in involved missions and data lengths between these models.

We have carefully read your review comments and respond to each of your feedback below. The black font is your review comments, the red font is the explanatory response, and the blue font is the revision in the revised manuscript.

**Detailed comments:**

**1.1 Comment 1**

1. Need to introduce DTU21, SS V32.1 in the abstract.

   Thanks for the suggestion. We have added the introduction of DTU21 and SS V32.1 in the abstract of the revised manuscript. The new abstract is shown below in blue text.

   **Abstract**

   Judging from the early release of the NSOAS22 model, there were some known issues, such as boundary connection problems in block-wise solutions and a relatively high noise level. By solving these problems, a new global marine gravity model NSOAS24 is derived based on sea surface slopes (SSS) from multi-satellite altimetry missions. Firstly, SSS and along-track deflections of vertical (DOV) are obtained by retracking, resampling, screening, differentiating, and filtering procedures on basis of altimeter waveforms and sea surface height measurements. Secondly, DOVs with a 1'x1' grid interval are further determined by the Green's function method, which applies directional gradients to constrain the surface, least-square fit to constrain noisy points, and tension constraints to smooth the field. Finally, the marine gravity anomaly is recovered from the gridded DOV according to the Laplace Equation. Among the entire processing procedures, accuracy improvements are expected for NSOAS24 model due to the following changes, e.g., supplementing recent mission observations and removing ancient mission data, optimizing the step size during the Green's function method, and special handling in near-shore areas. These optimizations effectively resolved the known issues of signal aliasing and the "hollow phenomenon" in coastal zones. The typical altimetry-derived marine gravity models are the DTU series released by the Technical University of Denmark and the S&S series released by the Scripps Institution of Oceanography (SIO), University of California

San Diego (UCSD). Their latest models, DTU21 and SS V32.1, were used for comparison and validation. Numerical verification was conducted in three experimental areas (Mariana Trench area, Mid-Atlantic Ridge area, Antarctic area, representing low, mid and high latitude zones) with DTU21, SS V32.1 and shipborne data. Taking NSOAS22 for contrast, NSOAS24 showed improvements of 1.2, 0.7, 1.0 mGal in 3 test areas by validating with SS V32.1, while declines of 0.6, 0.5, 0.3 mGal, and 0.2, 0.4, 0.3 mGal occurred in STD statistics with DTU21 and shipborne data. Finally, the NSOAS24 was assessed using two sets of shipborne data (the early NCEI dataset and the lately dataset from JAMSTEC, MGDS, FOCD, and SHOM) on global scale. Generally, NSOAS24(6.33 and 4.95 mGal) showed comparable accuracy level with DTU21 (6.20 and 4.71 mGal) and SS V32.1 (6.40 and 5.53 mGal), and better accuracy than NSOAS22 (6.64 mGal and 5.64 mGal). Besides, the new model is available at https://doi.org/10.5281/zenodo.12730119.

**1.2 Comment 2**

2. In Figure 1e, what are the stripes? Given CryoSat-2's inclination of 92°, one would expect N-S aligned stripes rather than the NW-SE pattern depicted.

   The stripes are caused by the satellite's trajectory. Yes, CryoSat-2's inclination is 92°, as shown in the figure below. This is the data distribution of a CryoSat-2 pass near the equator. It can be observed that the data distribution does not form perfectly N-S aligned stripe, but rather follows the NW-SE pattern depicted. Therefore, in Figure 1(e), such stripes are consistent with the trajectory distribution of the CryoSat-2 satellite.

[Figure]

Figure 1. A distribution map of CryoSat-2 data for a pass near the equator

**1.3 Comment 3**

3. Figure 2 caption: in dashed rectangular ◊ in dashed rectangle.

   Thank you for your reminder. All corrections have been made in the revised manuscript.

**1.4 Comment 4**

4. Equation 1: Ensure that every parameter in Equation 1, including 'N,' is defined within the text to avoid ambiguity.

   Thanks for the suggestion. All changes have been made in the revised manuscript according to your suggestions. The modifications in the revised manuscript are shown below in blue text.

$$\varepsilon^{\alpha} = -\frac{\partial N}{\partial s} = -\frac{\partial N}{\partial t} * \frac{\partial t}{\partial s} = -\frac{\partial N}{\partial t} * \frac{1}{v} \tag{1}$$

Here, $N$ is the height of the geoid, $s$ is the spherical distance, and $t$ is the observation time. The process for determining the linear velocity $v$ is as follows. Given a data point's latitude $\varphi$, we first convert the geodetic latitude

to geocentric latitude $\varphi_c$ by considering the Earth's flattening $e$. The formula is expressed as follows:

**1.5 Comment 5**

5. Line 276: what are the wavelengths used in the Parks-McClellan filter? Need to provide the wavelength range used for this study.

In marine gravity anomaly inversion, models derived from nadir altimeters achieve an accuracy of approximately 2–3 mGal and require low-pass filtering at wavelengths of at least 14 km to suppress short-wavelength noise amplified by differential derivative calculations (Sandwell et al., 2021). Therefore, we referenced this standard and applied a Parks-McClellan filter with a cutoff wavelength of 16 km. We have included a description of this section in the revised manuscript.

Reference

Sandwell, D.T., Harper, H., Tozer, B., Smith, W.H.F.: Gravity field recovery from geodetic altimeter missions. Adv. Space Res., 68, 1059–1072, https://doi.org/10.1016/j.asr.2019.09.011, 2021.

**1.6 Comment 6**

6. Line 281: what is sea surface topography? Is it mean dynamic topography? If so authors should to use this standardized term.

Thank you for your reminder. It is mean dynamic topography. All corrections have been made in the revised manuscript.

**1.7 Comment 7**

7. Figure 8 is very hard to read. Authors could possibly consider using histogram to compare the noise level and noise points alternatively.

Thanks for the suggestion. The histogram has been added according to your

suggestion.

[Figure]

**Figure 1.** Noise histogram with different step sizes (a: east-west component at step size 2; b: north-south component at step size 2; c: east-west component at step size 3; d: north-south component at step size 3)

**1.8 Comment 8**

8. Color bars (figures 7,8,10,11,14): The current colorbar make it challenging to distinguish between red and dark pink. Could consider color-blind-friendly colormaps such as turbo, or blue to red when showing differences in Figures 10 and 11.

Thanks for the suggestion. Based on your suggestions, we have re-plotted the figures using blue-to-red colormaps. The new figures are shown below.

[Figure]

**Figure 7.** Residual results of DOV components difference for different step size selections (a: east-west component at 2 steps; b: north-south component at 2 steps; c: east-west component at 3 steps; d: north-south component at 3 steps)

[Figure]

**Figure 8.** Noise analysis at different step sizes (a: east-west component at step size 2; b: north-south component at step size 2; c: east-west component at step size 3; d: north-south component at step size 3)

[Figure]

**Figure 10.** Location of distribution of nearshore areas disturbed by continental regions (a: east-west component; b: north-south component)

[Figure]

**Figure 11.** Difference in results in the nearshore area before and after the special processing
(a: east-west component; b: north-south component)

[Figure]

**Figure 14.** The residual gravity anomaly map of NSOAS24

**2 Reviewer #2**

(Comments to Author):

The manuscript presents a global gravity modal which includes latest unique HY-2 satellite data, besides the conventional altimetry missions, which were used for the development of DTU21 and Sandwell&Smith V32.1. From the analysis and comparison, it is clear that that the NSOAS24 model is an improvement against the predecessor model NSOAS22. The authors revised and optimized the data preprocessing, data editing, high-frequency noise reduction by mission-specific filter, and gridding procedure. All these steps lead to an improved Global marine gravity model compared to the predecessor model NSOAS22. The latest model NSOAS24 show comparable accuracy to DTU21 and Sandwell&Smith V32.1 model, which are well-known and internationally recognized. It is worth to note that in both NSOAS24 and V32.1 models, the marine gravity is inverted from the along track slope and Deflection of the Verticals. The model presented in this paper nearly outperform the V32.1 model globally and in the test regions as shown by the cross validation with ship borne gravity measurements. In general, the work presented is valuable for optimizing data processing approaches and improving the accuracy of marine gravity models. The manuscript can be accepted with a minor revision.

Dear reviewer:

The author's team would like to thank you very much for taking your valuable time to review the paper and for providing very valuable feedback and suggestions. We have carefully read your review comments and respond to each of your feedback below. The black font is your review comments, the red font is the explanatory response, and the blue font is the revision in the revised manuscript.

**Detailed comments:**

**2.1 Comment 1**

1. Line 102: As a model developed in 2024, why the Cryosat-2 data is not processed up to year 2024? 5 years of data missing there.

   Thanks for the suggestion. We utilized CryoSat-2 data spanning from 2010 to 2019 (a 9-year period), which was processed for collinearity and then consolidated into a single dataset. This dataset meets the requirements for data volume and spatial coverage density. The primary reason for not using data from 2020 onwards is the version updating of the CryoSat-2 data in 2020. This version change necessitates adjustments to our pre-processing methods, particularly with respect to the retracking procedure. In subsequent research, we plan to implement targeted pre-processing for the new data version and incorporate more up-to-date data in future studies to further enhance the model's accuracy.

**2.2 Comment 2**

2. Line 120: why in Figure 1 a,b,f,g visually look so much blurred/white?

   In Figure 1, we used the same GMT plotting script and color scale. The likely cause of this effect is the larger spacing between adjacent orbits of both HY-2A and Geosat. For example, during the Geodetic Mission (GM) phase, HY-2A's repeat cycle is 168 days, and the spacing between adjacent orbits is 17.31 km. By contrast, CryoSat-2's GM phase has a repeat cycle of 369 days, with an average ground trajectory spacing of only 7–8 km around the equator. This results in less dense data coverage compared to other satellites, causing Figures 1a, b, f, and g to appear visually more blurred or white.

[Figure]

**Figure 1.** Slope plot of satellite ascent/descent at different orbital inclinations (a, b, c, d, e represent ascending track slopes for HY-2A (99.3°), Geosat (108°), Jason-2 (66°), SARAL/AltiKa (98.55°), CryoSat-2 (92°) respectively; f, g, h, i, j represent descending track slopes for HY-2A, Geosat, Jason-2, SARAL/AltiKa, CryoSat-2 respectively).

**2.3 Comment 3**

3. Line 293: Matric singularity.

   Thanks for the suggestion. It has been changed in the revised manuscript. The modifications in the revised manuscript are as follows.

   Due to the requirements of the Green's function method regarding region size and data volume, the convergence of multiple vectors with different values at the same gridding points but with consistent directions can lead to matrix singularity.

**2.4 Comment 4**

4. Line 293: remove "By the way", replace with "It is worth to mention"

   Thanks for the suggestion. It has been changed in the revised manuscript. The modifications in the revised manuscript are as follows.

   It is worth to mention that the averaging step between each category was essential to address this issue.

**2.5 Comment 5**

5. Line 319: Section 4.3 Figure 5. It is good that the authors noticed the edge effect, boundary problems, however, it is not completely eliminated yet, as

shown in Figure 5b. What could be done to revise your approaches? Observe latitude 15 deg.

Thank you for your reminder. In fact, the image distortion occurred when converting from Word (89,183 KB) to PDF (7,932 KB). The original image underwent pixel compression. After downloading the PDF version used for the review (Figure 2), we noticed a dividing line in the center of the image (at latitude 15°) that was not present in the original Word document (Figure 3). We have included the original image below for reference. Our approach has effectively eliminated the edge effects. (Due to the upload requirement for PDF format, the image may still be distorted. Please download the original image in a ZIP file for better quality. (https://doi.org/10.5194/egusphere-2024-2307-AC2))

[Figure]

**Figure 2.** The distorted image in the PDF version.

[Figure]

**Figure 3.** Result of spline splicing method for DOV east-west components (a: original; b: new).

**2.6 Comment 6**

6. Line 376: That is not the only reason. The Area 3 is a region with high ocean dynamics in the Southern Oceans. Even the north-south DOV components degrade a lot, even though there are many many near Polar altimetry missions.

Thank you for your reminder. Based on your suggestions, we have made the corresponding revisions in the revised manuscript.

Satellites with lower inclinations, such as the Topex/Poseidon and Jason series, are unable to provide observations beyond 66°, and area 3, a region with high ocean dynamics in the Southern Oceans, exhibits a noticeable decline in DOV quality in high-latitude regions.

**2.7 Comment 7**

7. Line 396: Would the author ever considered filling the Land with DOV values from EGM2008 or other high resolution models like XGM2019, so that the boundary effects are mitigated?

Thanks for the suggestion. The construction of the NSOAS24 model utilized the remove-restore method, where EGM2008 served as the reference model. All calculations were based on the residual DOVs of the satellite altimeter data relative to the reference model. Even when land areas are filled with the

reference model, the residual DOVs over land remain zero during the remove step. Therefore, in the remove-restore method, model data cannot be filled into land areas for joint inversion.

**2.8 Comment 8**

8. Line 458: The reasoning for keeping Geosat data is addressed here. Geosat only cover up to latitude 72 deg, so it is not that high compared to others (SARAL, Cryosat-2 and HY2). The most important contribution is for the east-west components. In addition, Geosat GM data is only for 1.5 years, it may be dropped due to the lack of high accuracy. Do the authors have a gravity model predicted without Geosat data? How was the accuracy?

Thanks for the suggestion. The observation data from Geosat, with its unique 108° orbital inclination, includes 1.5 years of Geodetic Mission (GM) data from 1985/04 to 1986/11, and 3 years of Exact Repeat Mission (ERM) data from 1986/12 to 1990/01. We performed inversion of the global marine gravity anomaly field without using Geosat data, as shown in Figures 4 and 5. We compared the accuracy of gravity anomalies with and without Geosat data in three experimental areas, and the results are as follows. Table 1 presents the inversion results with and without Geosat data, compared with the DTU21 and V32.1 models. Table 2 shows the inversion results with and without Geosat data, compared with shipborne data. As seen in the comparison with DTU21 and V32.1, the differences between using and not using Geosat data are minimal. However, the results with Geosat data are slightly better than those without. When compared with shipborne data, the inversion results using Geosat data are superior, especially in the high-latitude region of experimental area 3. Compared to NCEI shipborne data, the inversion accuracy with Geosat data differs from that without Geosat data by 0.6 mGal, and compared to MGDS shipborne data, the difference in inversion accuracy is 0.3 mGal.

[Figure]

**Figure 4.** The residual gravity anomaly map of NSOAS24 (without Geosat).

**Figure 5.** The gravity anomaly map of NSOAS24 (without Geosat).

**Table 1.** Statistics of NSOAS24 (without Geosat) and NSOAS24 (with Geosat) against DTU21

and V32.1 (unit: mGal)

| Area | Model | Max | Min | Mean | STD |
|------|-------|-----|-----|------|-----|
| 1 | NSOAS24(without Geosat)-DTU21 | 219.44 | -286.78 | 0.02 | 2.93 |
| | NSOAS24(with Geosat)-DUT21 | 238.23 | -255.97 | 0.02 | 2.93 |
| | NSOAS22(without Geosat)-V32.1 | 165.17 | -274.09 | -0.02 | 1.96 |
| | NSOAS24(with Geosat)-V32.1 | 91.36 | -243.28 | -0.03 | 1.97 |
| 2 | NSOAS24(without Geosat)-DTU21 | 185.90 | -72.26 | -0.03 | 1.47 |
| | NSOAS24(with Geosat)-DUT21 | 104.50 | -72.45 | -0.03 | 1.46 |
| | NSOAS22(without Geosat)-V32.1 | 190.41 | -79.69 | -0.01 | 1.24 |
| | NSOAS24(with Geosat)-V32.1 | 109.01 | -101.60 | -0.01 | 1.23 |
| 3 | NSOAS24(without Geosat)-DTU21 | 195.52 | -223.08 | 0.02 | 6.02 |
| | NSOAS24(with Geosat)-DUT21 | 195.52 | -223.07 | 0.02 | 6.01 |
| | NSOAS22(without Geosat)-V32.1 | 305.44 | -188.37 | -0.08 | 3.64 |

NSOAS24(with Geosat)-V32.1    305.43  -188.36  -0.08  3.61

**Table 2.** Statistics on differences between altimeter-derived models and shipborne gravity data (unit: mGal)

| Area | Model | Ship-borne data | Max | Min | Mean | STD | Ship-borne data | Max | Min | Mean | STD |
|------|-------|-----------------|-----|-----|------|-----|-----------------|-----|-----|------|-----|
| 1 | NSOAS24(without Geosat) | NCEI | 43.26 | -42.43 | -0.66 | 5.64 | JAMSTEC | 37.90 | -44.25 | 1.14 | 5.04 |
| | NSOAS24(with Geosat) | | 37.07 | -41.06 | -0.68 | 5.60 | | 35.15 | -42.88 | 1.12 | 4.97 |
| | DTU21 | | 36.09 | -42.72 | -0.72 | 5.12 | | 24.90 | -26.20 | 0.56 | 4.37 |
| | V32.1 | | 54.68 | -68.25 | -0.68 | 5.07 | | 57.91 | -66.35 | 0.74 | 4.99 |
| | EGM2008 | | 15.00 | -15.00 | -0.61 | 5.70 | | 15.00 | -15.00 | 0.52 | 4.93 |
| 2 | NSOAS24(without Geosat) | NCEI | 28.71 | -27.07 | 3.24 | 7.23 | FOCD SHOM | 29.21 | -25.57 | 3.17 | 6.61 |
| | NSOAS24(with Geosat) | | 27.35 | -27.61 | 3.24 | 7.21 | | 26.67 | -25.17 | 3.17 | 6.60 |
| | DTU21 | | 23.96 | -22.73 | 3.16 | 7.17 | | 22.14 | -18.70 | 3.14 | 6.48 |
| | V32.1 | | 36.84 | -26.69 | 3.19 | 7.19 | | 29.45 | -19.20 | 3.16 | 6.45 |
| | EGM2008 | | 15.00 | 15.00 | 2.87 | 7.13 | | 15.00 | 15.00 | 2.78 | 6.74 |
| 3 | NSOAS24(without Geosat) | NCEI | 189.59 | -37.42 | 2.54 | 6.82 | MGDS | 70.59 | -68.45 | -0.06 | 6.12 |
| | NSOAS24(with Geosat) | | 189.58 | -38.36 | 2.56 | 6.21 | | 44.94 | -68.45 | -0.09 | 5.92 |
| | DTU21 | | 23.28 | -41.36 | 3.20 | 5.79 | | 44.68 | -58.59 | 0.16 | 5.83 |
| | V32.1 | | 279.57 | -142.13 | 2.64 | 7.79 | | 235.69 | -114.62 | 0.41 | 8.68 |
| | EGM2008 | | 15.00 | 15.00 | 2.42 | 6.28 | | 15.00 | 15.00 | -0.08 | 6.19 |

**2.9 Comment 9**

9. Line 506: "with a decline" replace with "with a reduction of 0.7 mGal"

Thanks for the suggestion. It has been changed in the revised manuscript. The modifications in the revised manuscript are as follows.

In general, NSOAS24 demonstrates accuracy comparable to DTU21 and V32.1. Compared to its predecessor, NSOAS24 shows a steady improvement in accuracy, with a reduction of ~0.7 mGal in standard deviations when compared with recent non-NCEI shipborne data.